# Carbon nanotube dry adhesives with temperature-enhanced adhesion over a large temperature range

Ming Xu[1,2], Feng Du[1], Sabyasachi Ganguli[3], Ajit Roy[3] & Liming Dai[1,4]

Conventional adhesives show a decrease in the adhesion force with increasing temperature due to thermally induced viscoelastic thinning and/or structural decomposition. Here, we report the counter-intuitive behaviour of carbon nanotube (CNT) dry adhesives that show a temperature-enhanced adhesion strength by over six-fold up to $143\,N\,cm^{-2}$ ($4\,mm \times 4\,mm$), among the strongest pure CNT dry adhesives, over a temperature range from $-196$ to $1,000\,°C$. This unusual adhesion behaviour leads to temperature-enhanced electrical and thermal transports, enabling the CNT dry adhesive for efficient electrical and thermal management when being used as a conductive double-sided sticky tape. With its intrinsic thermal stability, our CNT adhesive sustains many temperature transition cycles over a wide operation temperature range. We discover that a 'nano-interlock' adhesion mechanism is responsible for the adhesion behaviour, which could be applied to the development of various dry CNT adhesives with novel features.

[1] Center of Advanced Science and Engineering for Carbon (Case4Carbon), Department of Macromolecular Science and Engineering, Case Western Reserve University, 10900 Euclid Avenue, Cleveland, Ohio 44106, USA. [2] State Key Laboratory of Materials Processing and Die & Mold Technology, School of Materials Science and Engineering, Huazhong University of Science and Technology (HUST), Wuhan 430074, China. [3] Materials and Manufacturing Directorate, Air Force Research Laboratory, Dayton, Ohio 45433, USA. [4] BUCT-CWRU International Joint Laboratory, College of Energy, Beijing University of Chemical Technology (CWRU), Beijing 100029, China. Correspondence and requests for materials should be addressed to L.D. (email: Liming.Dai@case.edu).

Adhesion between different solids permeates all aspects of our daily lives. Depending on the nature of applications, adhesives are subjected to diverse environments, ranging from the polymer-based adhesives (for example, Scotch tape, Scotch super glue, 3M Super Sticky Post-it Notes) used at ambient temperature for daily essentials, through specially designed rubbery sealants for automobile and aerospace vehicles in sub-freezing space or polar regions, to ceramic or metallic permanent glues for some specific operation processes at high temperatures (for example, solar cell panels, space shuttle launching and landing). However, conventional adhesives often show structural or performance deterioration at extreme temperatures, which could cause catastrophes[1]. For example, for high-temperature applications (for example, $> 500\,°C$), ceramic adhesives and/or metal welding are normally considered since they can stand up to temperatures even over $1,000\,°C$ (refs 2,3). Nevertheless, interfacial debonding cannot be prevented due to the differential thermal expansions between the adhesive layer and target surfaces, especially over thermal transitions with a wide temperature range.

Resembling a gecko's adhesive foot hairs with additional superior mechanical, electrical, and thermal properties[4], carbon nanotubes (CNTs) with hierarchical structures (for example, vertically aligned) are ideal stable adhesives for thermal and/or electrical management at high temperatures. However, this possibility has yet to be realized largely due to difficulties in the design and development of CNTs with optimized hierarchical structures. A strong shear adhesion force of $\sim 100\,N\,cm^{-2}$—ten times that of the gecko foot adhesive (the strongest dry adhesion force among all climbing animals)—was previously reported for vertically aligned multi-walled CNTs (VA-MWNTs) against a glass substrate[4], where the van der Waals (vdW) force was demonstrated to be responsible for the strong adhesion. In this particular case, it was found that the entangled nanotube tops enhanced their intimate contacts with smooth surfaces, and hence resulted in strong van der Waals forces[4]. Although a hierarchical CNT/polymer fibrillar structure was proposed in a subsequent study to promote the rough surface adaptability[5], most reported carbon nanotube dry adhesives were tested against smooth surfaces (for example, glass plates, Si wafers) to enhance the vdW interaction[4,6-8]. Meanwhile, certain specially structured CNT adhesives with multi-functionalities have been explored. Examples include the patterning of VA-CNTs for self-cleaning[6], depositing entangled nanotube segments on the top of VA-CNTs for anisotropic adhesion force distribution[4], producing single-walled VA-CNTs to extend the operational temperature from room temperature (ca. $24\,°C$) to $\sim 200\,°C$ (ref. 7), and using VA-CNTs as electrically and thermally conducting dry adhesives at ambient temperature[8].

In this work, we achieve thermally enhanced CNT adhesives for high-temperature applications by rationally designing vertically aligned double-walled CNT (VA-DWNT) strands with bundled top nodes induced by plasma treatment. In contrast to our expectations, we find that the adhesion strength of our CNT dry adhesives on rough surfaces (for example, commercially-available copper foils, *vide infra*) increased with increasing temperature by over six-fold up to $143\,N\,cm^{-2}$ at high temperatures (for example, $1,000\,°C$), one of the highest values reported. This unusual adhesion behaviour can be rationalized by a 'nano-interlocking' adhesion mechanism, in which the plasma-induced CNT top nodes can easily penetrate into cavities of a naturally or temperature-induced rough surface, followed by temperature-induced deformation of the flexible double-walled CNT segments along the surface profile to form nano-interlocks (*vide infra*). The screw-like nano-interlocking interaction distinguishes the CNT dry adhesive reported in this study from other conventional adhesives, including conventional CNT dry adhesives[4,6-8] and polymeric dry adhesives[9-11]. Our CNT dry adhesives sustain many temperature transition cycles over a wide operational temperature range from $-196$ to $1,000\,°C$ with a high thermal stability and a temperature-enhanced adhesion strength up to $143\,N\,cm^{-2}$ (one of the highest adhesion strengths among all known pure CNT dry adhesives).

## Results

**Temperature-enhanced adhesion.** Figure 1a–d schematically shows the procedure for preparing our CNT dry adhesive (CNT length: $\sim 300–500\,\mu m$, diameter: $7–10\,nm$, Supplementary Figs 1 and 2) while details for the material preparation can be found in the Methods. Briefly, the VA-DWNT array was synthesized according to the published procedure[12], followed by well-controlled oxygen plasma etching[13,14] and argon annealing (Supplementary Figs 3 and 4) to remove the nonaligned nanotube segments, leading to the nanotube bundling and the top node formation (Fig. 1e,f). In previous studies, nonaligned nanotube top segments have been used to enhance the shear adhesion force (up to $\sim 100\,N\,cm^{-2}$) for VA-MWNT arrays against smooth surfaces (for example, glass plates, Si wafers)[4] whereas similar plasma-etched VA-MWNT arrays with bundled tops were found to show weak adhesions ($< 12\,N\,cm^{-2}$) on the smooth surfaces[8]. However, plasma etching has been previously demonstrated to provide the additional advantage of facilitating the hydrogen-bonding mediated interaction between nanotubes and hydrophilic surfaces. Unlike previous studies, the plasma-induced bundled top nodes were designed in this study to facilitate a node-guided penetration into the surface cavities (Fig. 1c,d), initiating and ensuring an intimate contact of the VA-DWNT adhesive with temperature-induced rough surfaces.

By sandwiching a free-standing film of the CNT dry adhesive ($4\,mm \times 4\,mm$, Fig. 1g) between two copper surfaces (Alfa Aesar, A Johnson Matthey Company) with the constituent CNTs normal to the target surface by finger pressing ($\sim 7\,N\,cm^{-2}$), we used a torch for heating to make a quick demonstration of the wide operation temperature range and measured the shear adhesion force using a digital spring balance (AWS H-110) (Fig. 1g–j, Methods, Supplementary Figs 5 and 6). The high melting temperature ($1,083\,°C$), together with its excellent thermal and electrical properties for thermal and electrical managements[15], makes the copper foil a model target surface for demonstrating CNT adhesion at high temperatures. Nevertheless, the methodology developed in this study is applicable to many other metallic and non-metallic substrates, including aluminum foil, glass plate, silicon wafer and polymer films (for example, a fluorinated ethylene propylene (FEP) film).

As seen in Fig. 1g, our newly developed CNT adhesive showed a room-temperature ($24.2\,°C$) adhesion strength of $\sim 37\,N\,cm^{-2}$, a value which is similar to that of a 3M double-side sticky tape (Methods). Upon continuous heating by a butane torch (Master Appliance MT-30 Table Top Self Igniting Microtorch with the flame temperature up to $1,970\,°C$), we found with surprise that the adhesion strength increased with increasing temperature from $\sim 37\,N\,cm^{-2}$ ($24.2\,°C$) through $\sim 60\,N\,cm^{-2}$ ($418.7\,°C$) to $\sim 124\,N\,cm^{-2}$ ($1,033\,°C$) (Fig. 1g–i). As far as we are aware, such a temperature-induced adhesion enhancement has never been reported for any existing adhesive materials, and the value of $124\,N\,cm^{-2}$ is among the highest adhesion strength for all known pure CNT dry adhesives (Supplementary Table 1). On the other hand, it was found that the room temperature adhesion strength remained almost unchanged ($\sim 34\,N\,cm^{-2}$) even when our CNT dry adhesive was cooled down near to the liquid nitrogen temperature ($-190.7\,°C$, Fig. 1j)—still outperforming

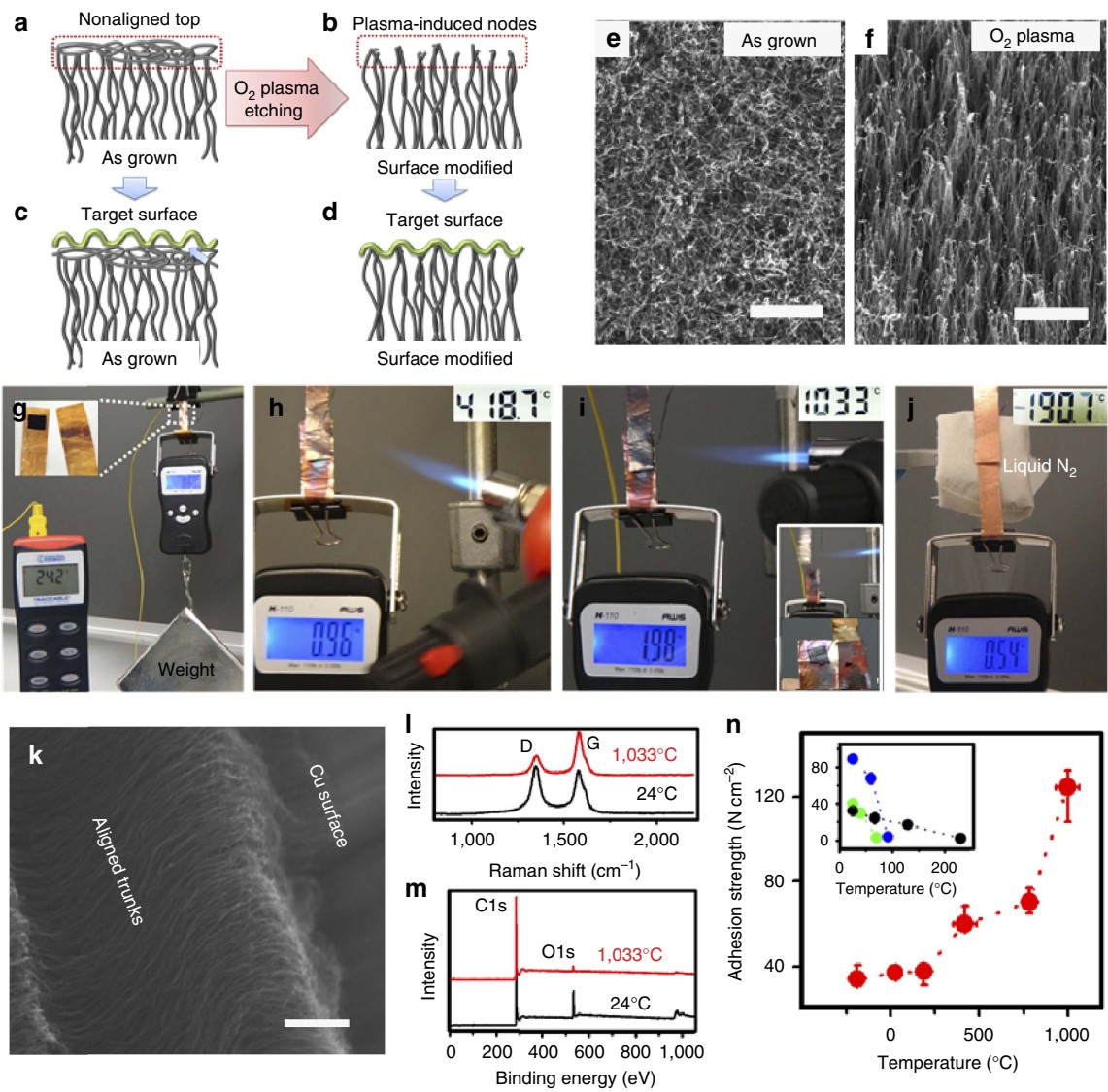

**Figure 1 | Adhesion enhancement of the carbon nanotube (CNT) adhesive at high and low temperatures.** (**a–d**) Diagram showing the preparation procedure of the CNT dry adhesive. (**e,f**) Top-view scanning electron microscopic (SEM) image of the rationally designed fibrillar adhering surface generated by plasma etching the nonaligned, entangled top nanotube segments on the as-grown VA-DWNT array (**e**) to form the bundled top nodes (**f**) (scale bars, 2 μm). (**g–j**) Digital photo images illustrating the adhesion force measurements at different temperatures. (**g**) room temperature measurement (24 °C), inset in the up-right showing the two copper foils to be 'glued' together by the VA-CNT dry adhesive within the black squared area. (**h,i**) Enhanced adhesion with increasing temperature up to 1,033 °C (inset at the bottom-right in **i** showing the debonding at 1,085 °C and the CNT dry adhesive after the high temperature test). (**j**) Liquid N₂ cooling ( − 190.7 °C) by using a thick piece of tissue papers pre-immersed in liquid N₂. (**k**) Side-view SEM image showing the structure integrity after the test at 1,033 °C (scale bar, 1 μm). (**l**) Raman spectra of the CNT dry adhesive after tests at 24 and 1,033 °C for comparison, and (**m**) the corresponding XPS spectra. (**n**) Temperature dependence of the adhesion force for the CNT dry adhesive (red curve). The inset shows temperature dependence of the adhesion forces for commercial adhesive tapes, including the Duro Liquid Super Glue (green curve), 3 M Polyimide Film Tape 5413 (black curve), and 3 M 410 M Double-Sided Masking Tape (blue curve) of the same size (4 mm × 4 mm). Each of the adhesion strength data points was averaged from five samples measured under the same thermal control conductions. The variation of the error bars is caused by the temperature fluctuation and the system error.

all conventional viscoelastic tapes (for example, 3M double-sided sticky tape), which would have become glassy and detached from the target surface in such a cold environment.

Figure 1k shows the retained vertical alignment of the nanotube trunks for a VA-DWNT dry adhesive even after being tested at 1,033 °C. The corresponding Raman spectra (Fig. 1l) and X-ray photoelectron spectroscopic (XPS) results (Fig. 1m) show an improvement of the nanotube structure after the high temperature testing due to a thermally induced increase in the graphitization degree with a concomitant loss of physically adsorbed oxygen molecules[16], as reflected by the increase in the

Raman G band relative to the D band and decrease in the XPS O 1s peak with respect to the XPS C 1s peak. This, together with the absence of catalyst residue from our nanotube materials (Fig. 1m; Supplementary Fig. 1), ensured the superior thermal stability even in air for our VA-DWNTs, consistent with the previous report[17].

To compare the temperature-enhanced adhesion of our CNT dry adhesive with commercially available adhesive tapes, we selected the Duro super glue ( ∼ 103 N cm⁻²) with a maximum operation temperature (MOT) of 93 °C, 3 M thermally conductive tape (32 N cm⁻², MOT = 260 °C), and 3 M double-side sticky tape ( ∼ 42 N cm⁻², MOT = 85 °C) of the same size

(4 mm × 4 mm) as the benchmarks since they have either a high adhesion force or high-temperature resistance, or both, among viscoelastic tapes. We found that these conventional adhesives showed a dramatic decrease in the adhesion force with increasing temperature (inset of Fig. 1n) due to the thermally induced viscoelastic thinning and/or structural decomposition. In contrast, our CNT dry adhesive exhibited a steadily increased adhesion force with increasing temperature up to ~1,000 °C (Fig. 1n). The above temperature dependence of the VA-DWNT adhesion demonstrated by torch heating was confirmed by more careful adhesion strength measurements on the CNT dry adhesive being thermally treated with an *in situ* (heated and adhesion measured in TA Instruments, RSA-G2 with Environmental Test Chamber) or *ex situ* (heated inside of a muffle furnace and cooled down to room temperature for the adhesion force measurements) temperature control (Supplementary Fig. 7). The *in situ* and *ex situ* temperature controlled measurements showed an agreement in the adhesion enhancement trend, implying that the irreversible adhesion enhancement with increasing temperature (≥ ~200 °C) has been reliably measured.

**Nano-interlocking dry adhesion mechanism.** The observed thermally induced adhesion enhancement (Fig. 1n) is the result of a nano-interlocking dry adhesion mechanism. To elucidate the nano-interlocking dry adhesion mechanism, we took scanning electron microscopy (SEM) images of the CNT adhesive surface after detachment (Fig. 2a). The top-view image in Fig. 2a shows uniformly distributed web-like domains with a center-to-center distance of 1–2 μm, which is significantly different from the randomly entangled top segments on the as-grown VA-DWNT array (Fig. 2d, cf. Fig. 1e). The enlarged view of individual web-like domains (Fig. 2c) clearly shows the web-like structure with CNT segments radiated out from the center. We further found that there were residual CNTs adhered onto the target copper surface in a web-like shape (Fig. 2b) with some of CNTs being overstretched into aligned nanotubes parallel to the shear direction. These CNT webs are distributed uniformly on the target surface, about 1–2 μm apart from each other, (Fig. 2b), which is a close replication of the CNT coils shown in Fig. 2a. The formation of these web-like contacts is further revealed by SEM examination of the structural deformation of an individual CNT bundle as a model under the normal loading (Fig. 2e–j). Figure 2e,h shows that the adhesion process commences from the point-contact between the CNT node and the target surface. Under an appropriate pre-loading force (>25 N cm$^{-2}$, Supplementary Fig. 8), the VA-DWNT bundle starts to deform (Fig. 2f,i) and finally collapses through the web-like deformation into the line contacts with the target surface around the node (Fig. 2g,j). Thus, it is likely that the node of the VA-DWNT bundle penetrates into a surface cavity during the contact process, from which CNTs in the bundle radiate out through the web-like deformation with some of them being overstretched into aligned nanotubes parallel to the shear direction upon shear (Fig. 2b,g). The screw-like, node-guided penetration of CNTs into the surface cavities and the associated side contacts between the nanotubes and cavities after the web-like deformation are evident in the SEM images shown in Fig. 2a,c. The observed web-like collapsing of the CNT bundle can be schematically depicted in Fig. 2h–j, which under normal compression collapses into a web-like structure to allow for line (side) contacts with the target surface around the node. Therefore, the node-guided contact mode, in conjunction with the web-like collapsing described above, provides synergistic effects for the use of our CNT dry adhesives on rough surfaces. The wall number of CNTs is a crucial factor to control the

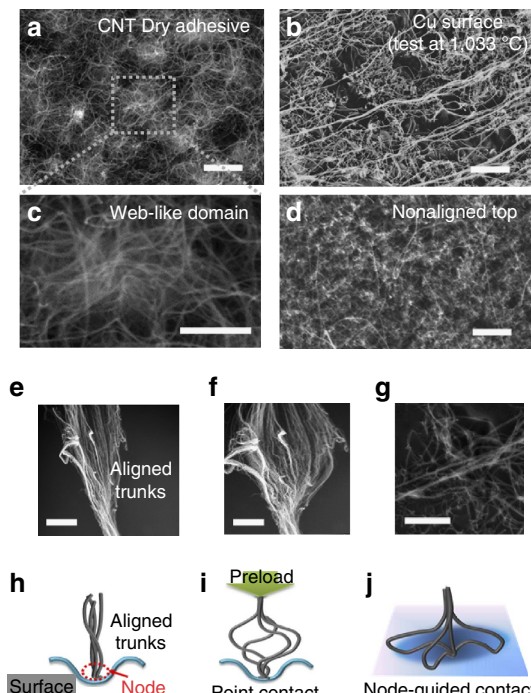

**Figure 2 | Model of the node-guided contacting mode.** (**a,b**) SEM images for the CNT adhesive after detachment, showing web-like CNT segments on the top of the CNT adhesive (**a**) and the copper surface (**b**), respectively, after testing at 1,033 °C (scale bar, 1 μm). (**c**) A higher magnification view of the CNT web selected by the dotted box in **a** (scale bar, 0.5 μm). (**d**) nonaligned top CNT segments of the pristine vertically aligned double-walled CNT (VA-DWNT) array under the same magnification as **b** for comparison (scale bar, 1 μm). (**e–g**) SEM images of the CNT bundle deformation under normal preloading to induce line contacts with the target surface through the web-like deformation (scale bar: (**e**) 20 μm; (**f**) 20 μm; (**g**) 1 μm). The individual CNT bundle was drawn and twisted by a pair of tweezers from the CNT dry adhesive to mimic the plasma-induced bundles with top nodes. Due to technical difficulties, the actual size of the individual CNT bundle thus produced is at least 20 times that of the plasma-induced bundles. (**h–j**) The corresponding schematic depictions for **e–g**.

nanotube elasticity for plasma-induced bundling and node formation. Thus, the VA-DWNTs were chosen for their proper flexbility and good thermal stability associated with high purity[17]. The plasma-induced formation of the VA-DWNT bundles with a top node could overcome the over-softness of the individual DWNTs to ensure effective penetration into the cavities on rough surfaces and to allow for intimate point and side contacts with the target surface, and hence the maximized adhesion for the CNT dry adhesive[4,7]. As shown in Fig. 1c, however, the nonaligned top on the as-grown VA-DWNT array would prevent the CNTs from an intimate contact with a rough surface, though the top nonaligned nanotube segments on a VA-MWNT array have been reported to play an essential role in enhancing the CNT adhesion on a smooth surface (for example, glass plate) through shear-induced line contacts to enlarge the effective contact area, and hence the enhanced vdW force[4,5].

**Adhesion enhancement model.** To more quantitatively study the interlock adhesion mechanism and the associated thermally induced adhesion enhancement, we replot the data from the *ex situ* measurements shown in Fig. 1n, along with the corresponding plot of the enhancement factor (defined as the

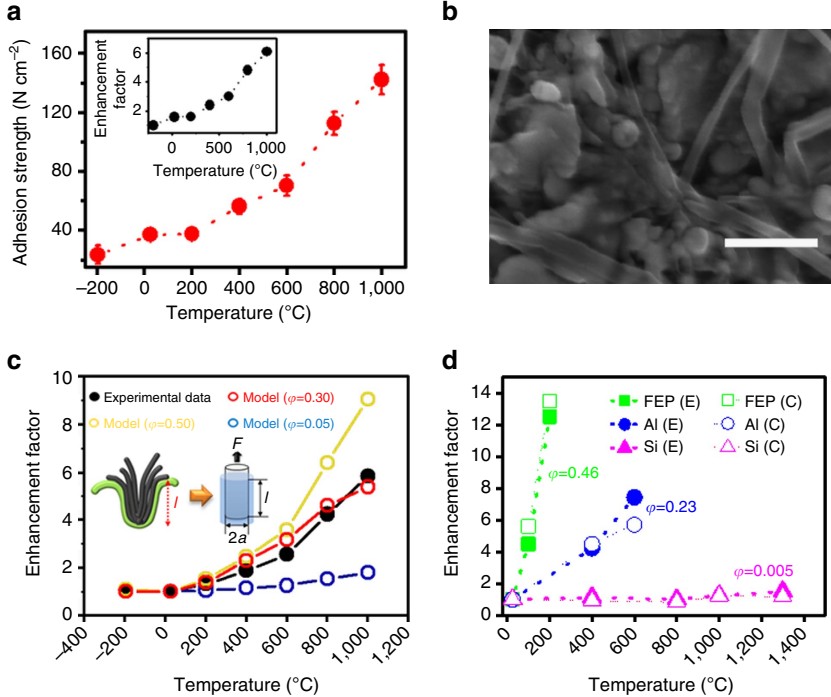

**Figure 3 | Adhesion enhancement model.** (**a**) Temperature dependence of the adhesion force for the CNT dry adhesive. Inset shows the enhancement factor versus temperature. (**b**) SEM images for the residual CNTs on the detached copper surface after testing at 1,000 °C, showing the temperature-induced screw-like nano-interlocking (scale bar, 2 µm). (**c**) Calculated temperature-dependence of the enhancement factor (red, blue and orange circles) (cf. equation (1)) along with the corresponding experimental data (black dots) for comparison. (**d**) Verification of the nano-interlocking model for different target surfaces. Each of the adhesion strength data points was averaged from five samples measured under the same *ex situ* thermal control conductions.

normalization of adhesion strength at a specific temperature to the corresponding adhesion strength at room temperature), as a function of temperature in Fig. 3. As seen in the inset of Fig. 3a, the CNT adhesion increases by about six times ($\sim 6.2 \times$) with temperatures over $-196$ to $\sim 1,000$ °C. However, either the vdW force dominated CNT adhesion[4] or the viscoelastic interactions between the CNTs should be insensitive to the temperature variation, as demonstrated previously[17]. Clearly, therefore, an unknown yet decisive factor, in addition to the vdW and viscoelastic interactions, is regulating the observed temperature influence on the adhesion performance of the CNT adhesives.

Our SEM images taken from the copper foil under annealing at different temperatures show that surface segregation occurred upon heating the relatively smooth copper foil above $\sim 200$ °C, leading to an increase in the surface roughness with increasing temperature (Supplementary Figs 9 and 10). It could guide the collapsed CNT segments around the node (cf. Fig. 3b) to penetrate further into the temperature-induced irregular profiles to mechanically lock them into the surface, which explained the enhanced adhesion under higher preloading by pushing more CNT (bundle) "screws" into deeper surface cavities to produce higher vdW forces for stronger surface fastening (Fig. 3b; Supplementary Fig. 11).

To verify the enhanced adhesion through this CNT nano-interlocking mechanism, we estimated the extra work to pull the CNT bundles out of pits. The VA-DWNT bundle was modelled as an elastic strand to be embedded in a surface asperity, which was assumed as a cylindrical rigid hole with depth of $l$ and radius of $a$ for simplicity, and subjected to pull-out force ($F$) (inset in Fig. 3c, cf. Supplementary Fig. 12). While the detailed derivation is given in Supplementary Equations 1–9 (Supplementary Discussions), equation (1) gives the

enhancement factor at a specific temperature ($T$) as a function of the pull-out work associated with the asperities at $T$ defined by $l(T)$ and the fractional area of holes with CNT strand embedded in per unit area of surface, $\varphi$ (where, $\varphi = n\pi a^2$, ranging from 0 to 1, and there are $n$ holes per unit area of the surface):[18]

$$\frac{G_a(T)}{G_a} = 1 + \left[4\left(\frac{l(T)}{a}\right) - 1\right]\varphi \qquad (1)$$

The above equation illustrates that the screw-like enhancement mechanism at high temperatures produces deeper holes, leading to a deeper penetration of CNTs into the holes for stronger surface fastening. Equation (1) reveals that if once the depth $l(T)$ is increased to greater than one-fourth (1/4) of the radius of the hole ($a$), the screw-like enhancement makes the contribution to the extra energy for detachment. The embedded length $l(T)$ was approximated by using the roughness values ($R_q$) from the atomic force microscopic (AFM) investigation on the target copper surface at different temperatures (cf. Supplementary Fig. 10; Supplementary Table 2) and the radius of the hole was $\sim 60$ nm from SEM observation. We estimated the enhancement factors $\frac{G_a(T)}{G_a}$ at different temperatures from equation (1) and found that it agreed well with the experimental data while $\varphi$ was taken as 0.30 (red dots in Fig. 3c, along with the model calculation by using $\varphi = 0.05$ and 0.50 for comparison). This result means that about 30% of the surface area contributes to the nano-interlocking extra energy in addition to the vdW contact.

Owing to the generic nature of the thermally induced surface roughness, the concept of the node-guided CNT nano-interlocking reported here could be regarded as a general strategy for the development of thermally enhanced CNT dry adhesives for a variety of target surfaces, ranging from polymer films to metal foils, over a wide range of operation temperatures

(Supplementary Fig. 13). Indeed, Fig. 3d shows similar thermally induced adhesion enhancements for our CNT dry adhesives against a FEP film (American Durafilm, MOT = 204 °C (ref. 19)) and aluminum foil (Al, Fisher Scientific, MOT = 600 °C), but not against a silicon wafer (Silicon Quest International, MOT = 1,300 °C). As expected, the temperature-invariant adhesion seen for the silicon plate in Fig. 3d suggests that the temperature-insensitive, hard and smooth silicon surface could not support the formation of the thermally induced interlocking structure with a very small value of 0.005 for $\varphi$. For the FEP and Al surfaces, a good agreement between the model fit with equation (1) and experimental data was obtained when $\varphi$ is 0.46 and 0.23, respectively, indicating that 46% and 23% of the surface areas contribute to the nano-interlocking extra work in addition to the vdW force with the FEP and Al surface.

**Electrical and thermal managements.** As can be seen from the above discussions, our CNT dry adhesives have potential for various practical applications. In this context, the CNT dry adhesive was further subjected to electrical (Signatone S-1160 Probe Station) and thermal transport testing (LFA 457, Netzsch Laser flash) in the adhesive assembly (Supplementary Fig. 14). It was found that our CNT adhesive exhibited steadily increasing electrical conductivity ($10^3$–$10^4$ S cm$^{-1}$) and thermal diffusivity (0.1–0.25 mm$^2$ s$^{-1}$, albeit relatively low in respect to the reported data for a VA-MWNT array[20,21]) with increasing temperature (adhesion strength) up to 1,000 °C (Fig. 4a,b, cf. Fig. 3). The one-order increase in the electrical conductivity is most probably associated with the thermally induced nano-interlocking, which enhances the CNT-target surface (that is, copper) contact, and hence the enhanced electrical conductivity and adhesion force

(Fig. 4b). Compared with the temperature-enhanced electrical conductivity, the lesser increase in the thermal diffusivity (less than 2-fold) at high temperatures (Fig. 4b) indicates that some more stringent requirements on the CNT-target surface contact have to be met for efficient thermal transport[21]. We also investigated the long-term stability for the electrical transport during repeated temperature transition cycles over a wide temperature range from −196 to 1,000 °C. As seen in Fig. 4, our CNT dry adhesive could not only retain the adhesion force (Fig. 4b) but also exhibit increased charge-carry capacities over repeated heating (1,000 °C)–cooling (−196 °C)–heating (1,000 °C) cycles (Fig. 4c) due, once again, to the increased conductivity by the thermally enhanced CNT-copper contact. When performing the (1,000 °C)–cooling (25 °C)–heating (1,000 °C) cycles, we found that the CNT dry adhesive initially underwent the step-by-step increase in conductivity (inset of Fig. 4c, ($g\rightarrow n$)) associated with an irreversible enhancement in the adhesion caused by the temperature-enhanced surface roughness. Thereafter, the step-by-step conductivity increase disappeared and a reversible sharp increase in current was observed upon heating up to about 65 cycles (Fig. 4c) due to the increased electrical conductivity of the nanotube array with increasing temperature (Supplementary Fig. 15), as is the case for VA-MWNTs[22]. We believed that the reversible fluctuation of the current at the cooling (−196 °C) — heating (25 °C) — cooling (−196 °C) cycles (inset of Fig. 4c, ($a\rightarrow f$)) was also caused by the temperature-dependence of conductivity for the nanotube array. An ohmic contact between the Cu and CNT dry adhesive was also evident by linear $I$–$V$ curves measured at different temperatures (Supplementary Fig. 16). For conventional adhesives, long term exposure to temperature variations often causes detachment even within their operation temperature ranges due to the thermal

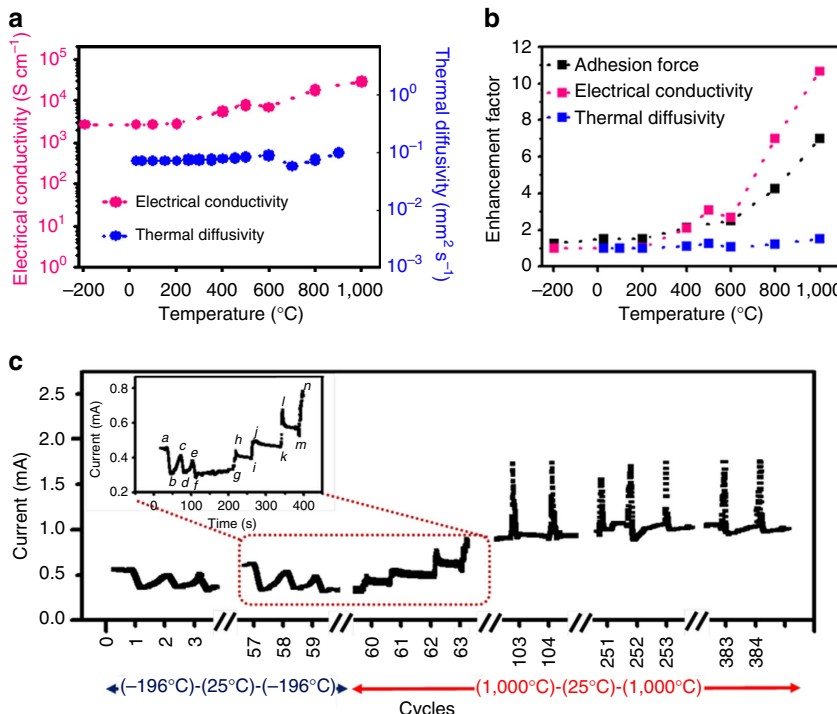

**Figure 4 | Electrical and thermal properties on the temperature transformation.** (**a**) Temperature dependence of the electrical conductivity and thermal diffusivity of the CNT adhesive from −196 to 1,000 °C by testing the Cu/CNT/Cu structure (cf. Supplementary Fig. 15). (**b**) Correlation between the electrical conductivity, thermal diffusivity and the mechanical adhesion enhancements with increasing temperature. (**c**) Stability of the CNT adhesive reflected by the electrical current (at 1 mV) going through the Cu/CNT/Cu structure under about 400 repeated temperature transition cycles. Inset shows an enlarged view of the heating and cooling transitions from 57th to 63rd cycles: a (25 °C) → b (−196 °C) → c (25 °C) → d (−196 °C) → e (25 °C) → f (−196 °C) → g (−196 °C) → h (1,000 °C) → i (1,000 °C) → j (1,000 °C) → k (25 °C) → l (1,000 °C) → m (25 °C) → n (1,000 °C).

expansion mismatch between the adhesive layer and the target surface(s).

In addition to the thermally induced nano-interlocking interactions described above, similar adhesion behaviours were observed for our newly developed CNT dry adhesives against various naturally rough surfaces, including rough metal foils, plastic films, wood pieces, paper sheets, and even painted walls, at ambient temperature (Supplementary Figs 17 and 18 and associated Supplementary Movies 1 and 2, see also Supplementary Table 3). These results clearly indicate the potential of our CNT dry adhesives for room temperature applications with a variety of rough surfaces, apart from the low- and high-temperature applications described above. Thus, the newly discovered node-guided nano-interlocking adhesion mechanism can be applied for the development of high-performance CNT dry adhesives for a large variety of applications, ranging from adhesion to electrical and thermal management, over wide operation temperature ranges from $-196$ to $1{,}000\,^{\circ}\mathrm{C}$.

## Methods

**Double-walled CNT array synthesis and fabrication.** The vertically-aligned carbon nanotube (VA-CNT) arrays were synthesized by low pressure chemical vapor deposition (CVD) on $4 \times 4\,\mathrm{mm}^2$ $SiO_2$ (400 nm)/Si wafers[12]. To start with, a 3-nm thick Fe layer was sputter coated on the wafers after the deposition of a 10-nm Al film. The catalyst coated substrate was then firstly inserted into a quartz tube furnace and heated up to $650\,^{\circ}\mathrm{C}$ in air. This was followed by pumping the furnace chamber to a pressure less than 30 m torr. The aligned CNT arrays were grown by flowing a mixture gas of 40v/v% Ar, 30v/v% $H_2$, 30v/v% $C_2H_2$ at $750\,^{\circ}\mathrm{C}$ under 200 torr for 10–20 min. The resulting VA-CNT arrays were examined on scanning electron microscope (SEM, Hitachi S-4500), and VA-CNT arrays with length around 300–500 μm long were selected for dry adhesive testing in this experiment. Then, a piece of glass slide was used to gently push the VA-CNT array from its side in the direction parallel to the substrate. By so doing, the VA-CNT was easily removed whilst the structural integrity was maintained.

**Adhesion force measurements.** By using a digital spring balance (AWS H-110), the adhesion force was measured at the shear-direction by sandwiching the CNT dry adhesive (4 mm × 4 mm) between two copper surfaces (Alfa Aesar, A Johnson Matthey) by finger pressing (preloading force of $\sim 7\,\mathrm{N\,cm}^{-2}$) (Supplementary Fig. 5a). For a quick demonstration of the adhesion at high temperatures, the CNT dry adhesive was directly heated up to $>1{,}000\,^{\circ}\mathrm{C}$ by a butane torch (Master Appliance MT-30 Table Top Self Igniting Microtorch with the flame temperature up to $1{,}970\,^{\circ}\mathrm{C}$) within 15 s (Supplementary Fig. 5b). Even in the air, the CNT dry adhesive was sufficiently stable at temperatures for the test[17]. With the adhesion force increased with increasing temperature, the weight was increased by holding the hook of the spring balance and drawing down until debonding happened (Supplementary Fig. 5b). A digital camera (Sony DSC-TX100) was used to capture the temperature and adhesion force. As shown in Supplementary Fig. 5b, the CNT dry adhesive exhibited the held weight of 1.98 kg (that is, $124\,\mathrm{N\,cm}^{-2}$) at $1{,}033\,^{\circ}\mathrm{C}$. Finally, the adhesion failure happened at $1{,}085\,^{\circ}\mathrm{C}$ (the melting point of copper, inset of Fig. 1i). For a quick demonstration of the low temperature test at $-190.7\,^{\circ}\mathrm{C}$, we used a thick piece of tissue papers pre-immersed in Liquid $N_2$ to quickly cool down the CNT adhesive (Supplementary Fig. 5c). The adhesion force measurements at both the high and low temperatures were repeated by heating the CNT dry adhesive inside of a muffle furnace (LABEC 240 V/2,000 W, Ceramic Engineering) and cooling in a liquid nitrogen container, respectively (see text). Plasma treatments were performed on a customer-built plasma reactor powered by commercial plasma generator (AGO201 HV, T&C Power Conversion, Inc.)[14]. We used the $O_2$-plasma etching to treat both the top and bottom surfaces of the free-standing CNT film to create the similar bundled surface topography with nodes on the both sides.

**Adhesion strength calculation.** The adhesion strength was calculated by dividing the value of the adhesion force by the CNT adhesive size (that is, the contact area). Supplementary Figure 6 shows the relationship between the adhesion strength and the contact side length of square CNT adhesive samples (that is, the CNT adhesive contact area). As can be seen in Supplementary Fig. 6, the adhesion strength keeps unchanged with changing the adhesive contact area up to 1 cm × 1 cm, followed by a slight decrease in the adhesion strength with further increasing the adhesive contact area up to 2.5 cm × 2.5 cm, presumably because the contact is normally poorer with a larger contact area, as is the case for all sorts of dry adhesives.

**SEM observation.** For structure observation, SEM images were taken using a Hitachi S-4500 instrument to observe the surface morphologies of the CNT arrays and target surfaces.

**TEM observation.** For nanotube structural characterization, transmission electron microscopy images were taken using a JEOL JEM-2000FX instrument.

**Raman.** Raman spectroscopy was performed on a Thermo-Electron Raman spectrometer with 532-nm excitation wavelength.

**X-ray photoelectron spectroscopic.** XPS measurements were carried out on a VG Microtech ESCA 2000 using a monochromic Al X-ray source (97.9 W, 93.9 eV).

**Electrical and thermal property measurements.** Electrical conductivity was measured on Signatone S-1160 Probe Station while thermal diffusivity was measured on LFA 457, Netzsch Laser flash.

**Atomic force microscopy.** AFM investigation was carried out on a Model of 5420 from Agilent Technologies.

**Surface roughness and morphology measurements.** Surface roughness and morphology were measured by AFM in a non-contact mode. To remove the contamination, the samples were cleaned by ethanol and dried before measurements $1 \times 1\,\mu\mathrm{m}$ area AFM images were used for analysis. The root mean squared roughness ($R_q$) as $R_q = \sqrt{\frac{1}{n}\sum_{i=1}^{n} y_i^2}$ was adopted to distinguish the roughness features of various target surfaces. Five tests were conducted on different areas to get average values.

**Data availability.** All the data that support the findings of this study are available from the corresponding author upon reasonable request.

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

## Acknowledgements

This work was support by AFOSR (FA9550-12-1-0037) and NSF (CMMI-1400274). This work was jointly supported by the National Thousand Talents Plan of China, the National Natural Science Foundation of China (51402117, 51572095), State Key Laboratory of Materials Processing and Die & Mould Technology, Fundamental Research Funds for the Central Universities, and Shenzhen Basic Research Project (JCYJ20140903171444756).

## Author contributions

M.X. and L.D. initiated the idea and designed the experiments; M.X., F.D. and S.G. performed the experiments. M.X., L.D. and A.R. analysed the data and co-wrote the paper.

## Additional information

**Competing financial interests:** The authors declare no competing financial interests.

