## [Peer Review File · Nature Communications]

Reviewers' comments:

Reviewer #1 (Remarks to the Author):

The authors describe the behaviour of CNT based dry adhesives that exhibit temperature-enhanced adhesion strength by over six fold, claiming it is the highest adhesion strength achieved among all known dry adhesives over a broad temperature range. They further suggest that such adhesion behaviour led to thermally enhanced electrical and thermal transport through the CNT dry adhesive, allowing for the these to be used as conductive double-sided sticky tapes for efficient electrical/thermal 'managements'.

The paper is not too well written or structured with numerous grammatical errors and repetitions within the text restating the same information. The paper requires major revisions to be carried out and important points to be addressed prior to recommending this for publication. Especially, high-standard has to be assured when submitting to Nature Communications which publish high-quality papers that represent significant scientific advances...

1. The most important point to address is that it is not really clear that it is surface roughness that is helping the bonding or whether some electrical bonding is going on. I don't see any atomic force microscopy images of roughened surface at those temps.

Why does the surface get rougher? Grains can grow but that does not make the surface considerably rougher. The authors should show data for adhesion to other surfaces to prove it is not a connection between copper and the tubes only that gives the enhancements.

I am not convinced that the authors have really explained why it has a higher adhesion and this is indispensable to support the central claims of this paper...

2. English and Grammar: many improvements and corrections are needed but predominantly visible is the typo in the title which should say 'range' rather than 'rang'.

3. In the abstract 'folds' need to be corrected to 'fold' and 'behavior' should be corrected throughout to 'behaviour'. Correct the following sentence to: '...the highest adhesion strength among all known dry adhesives) over a temperature range of -196 ~ 1000 °C.' Also, correct the '...this unusual behaviour led...' to: '... This unusual adhesion behaviour leads...'. Correct 'double-side' to 'double-sided'.

4. On Pg3, '...an extremely high temperature (e.g., 1000 °C...). The melting point of Cu is 1085°C so the copper is nearly molten!?

5. On Pg4, '...the highest adhesion strength among all known CNT dry adhesives.'. Isn't this just a repeat of what was said above?

6. On Pg6, '.... The in-situ and ex-situ temperature controlled measurements shows agreements in the adhesion enhancement trend, implying the irreversible....' Is it really irreversible? Explain and prove.

7. Pg6, Needs to be corrected to: '...was the result from a newly-discovered "nano-interlock" dry adhesion mechanism. To elucidate the "nano-interlock" dry adhesion mechanism, we took SEM images of the CNT...'

8. Pg7, Correct 'stability'...

9. Pg8, 'Clearly, therefore, an unknown yet decisive factor, in additional to the vdW and viscoelastic interactions...'. Correct this to: 'Clearly, therefore, an unknown yet decisive factor, in addition to the

vdW and viscoelastic interactions...'

10. Pg10, Correct 'As can be seen from the above discussion, our CNT dry adhesives hold potentials for various..' to: '...As can be seen from the above discussion, our CNT dry adhesives have potential for various...'

11. Pg11, Should be 'data' not 'date'?

Reviewer #2 (Remarks to the Author):

In this paper the authors measure the temperature dependence of the adhesion of carbon nanotube dry adhesives. They find that the adhesion of these dry adhesives shows an increase in strength with temperature approaching $\sim 140 \text{ N/cm}^2$. They propose an "interlock" model to account for this behavior. The ideas are novel but the quality of the data and the presentation needs some work. These are outlined below. I support publication of this manuscript after a few comments are addressed.

1. Did the authors try heating up and cooling and then measuring the adhesion after cycling the temperature? Does this annealing of the contact change the adhesive properties?
2. Grammatical correction on page 5 "consistent with the previous report"
3. Grammatical correction on page 6 "furnace and cooled down"
4. Grammatical correction on page 6 " (Fig. 1n) resulted from a newly-discovered"
5. Typo on page 7 " thermal stability"
6. Typo on title " Temperature Range"
7. Please explain what the error bars in the figures represent. How were they calculated? Figure 1n has error bars and also Figures S4f. For the case of Fig. S4F are those variations between samples and how many samples?
8. Typo on Supp Fig. S14 caption " ambient temperature"
9. Typo on Supp Fig. S14 caption " ambient temperature"
10. Typo on Supp Fig. S15 caption " weight was held on the painted wall"
11. Typo on Supp Fig. S15 caption " after deliberately taking each"
12. Please specify how the adhesion strength was calculated from the measurement? Also, how did the adhesion strength vary with contact area.
13. The images in Fig. S4a - e look almost identical. I don't see any nanotube bundling as in Figure 1 f. Why is this?

Reviewer #3 (Remarks to the Author):

The authors present a carbon nanotube based dry adhesive that increases in adhesion strength as temperature increases. A mechanism is proposed where the nanotubes 'screw' into naturally or temperature-induced rough surfaces enabling the increase in adhesion as temperature increases. The work goes on to show that these materials and mechanisms can also be useful for electrical and thermal conduction at interfaces.

Although interesting, the experimental results and procedures, novelty, and claims do not match the requirements for publication in Nature Communications. Specifically:

- 1) The experimental procedure for measuring the adhesive strength is not standardized and does not appear to be precise. Specifically, the adhesives were evaluated by pulling on a force scale as a butane torch was aimed at the interface. Without a controlled application of load and heat the results could contain artifacts (from rate effects, non-controlled loading conditions, localization of heat, etc.)

and may not represent the actual interfacial strength.

2) The work claims to present the highest adhesion strength for any carbon nanotube adhesive, and more broadly, any dry adhesive. This is inaccurate. Measurements on the adhesion strength of a single gecko seta demonstrated values on the order of 400 N/cm^2 (Autumn et al. Nature 2000). Therefore, this is not the strongest dry adhesive. Additionally, it is not the strongest carbon nanotube based adhesive, as Rong et al. (Adv. Mat. 2014) demonstrated a CNT adhesive with a shear adhesion strength of $185 \pm 50 \text{ N/cm}^2$.

3) The "nano-interlock" mechanism is not sufficiently supported. The schematics are clear enough to show the reader the hypothesis, however it is unclear how the images relate to the schematics. Without further evidence the current presentation is not convincing. Additionally, it is not clear how this 'could be applied to the development of various new dry adhesives and novel features.' It appears to be specific to this material set under the examined conditions.

4) The introduction discusses dry adhesives inspired by the gecko. One of the more intriguing aspects of the gecko is the highly reversible nature of the adhesive, where the adhesive can be applied and removed over many cycles. The CNT adhesive in this work is not demonstrated over multiple loading and release cycles, and upon removal it appears to damage the surfaces to which it was attached, as demonstrated in Figure S15. This adhesive is therefore more suitable as a permanent adhesive, and should be compared to other permanent adhesives instead of the pressure sensitive varieties such as 3M's double sided and thermally conductive tape. A more relevant and compelling comparison would be against permanent adhesives. In this case, the CNT adhesive is not as strong as established adhesive technologies such as epoxies which can obtain strengths on the order of 1500 N/cm^2 .

5) The work appears very similar to previous publications from the research group, specifically Qu, L. , Dai, L. Adv. Mat 2007 ("Gecko-Foot-Mimetic Aligned Single-Walled Carbon Nanotube Dry Adhesives with Unique Electrical and Thermal Properties"). This reduces the novelty of the current contribution. Furthermore, these related works raise the question on the necessity of the multiwall carbon nanotubes (VA-MCNTs) presented in this work relative to singlewall carbon nanotubes (VA-SWNT) presented in the previous work. In the author's previous work, adhesion strengths of $15\text{-}30 \text{ N/cm}^2$ were achieved with the SW-CNT at room temperature. This is similar to the $30\text{-}40 \text{ N/cm}^2$ reported here for VA-MCNTs. This leaves the reader wondering if the SW-CNTs have the same temperature dependent properties. In the larger scope, clarifying which CNT structure is most appropriate could help inform future research.

Point-To-Point Responses To Reviewers' Comments (NCOMMS-16-10739-R1):

Reviewer #1 (Remarks to the Author):

The authors describe the behaviour of CNT based dry adhesives that exhibit temperature-enhanced adhesion strength by over six fold, claiming it is the highest adhesion strength achieved among all known dry adhesives over a broad temperature range. They further suggest that such adhesion behaviour led to thermally enhanced electrical and thermal transport through the CNT dry adhesive, allowing for the these to be used as conductive double-sided sticky tapes for efficient electrical/thermal 'managements'.

The paper is not too well written or structured with numerous grammatical errors and repetitions within the text restating the same information. The paper requires major revisions to be carried out and important points to be addressed prior to recommending this for publication. Especially, high-standard has to be assured when submitting to Nature Communications which publish high-quality papers that represent significant scientific advances...

Response: We thank the Reviewer for his/her kind comments and the nice recommendation for revision.

1. The most important point to address is that it is not really clear that it is surface roughness that is helping the bonding or whether some electrical bonding is going on. I don't see any atomic force microscopy images of roughened surface at those temps. Why does the surface get rougher? Grains can grow but that does not make the surface considerably rougher. The authors should show data for adhesion to other surfaces to prove it is not a connection between copper and the tubes only that gives the enhancements.

I am not convinced that the authors have really explained why it has a higher adhesion and this is indispensable to support the central claims of this paper...

Response: *We thank the Reviewer for his/her comments. The Reviewer might have overlooked the AFM images in our original manuscript (Fig. S8). As shown in both SEM images (Fig. S7) and AFM images (Fig. S8) in our original Supporting Information, the roughness of copper surfaces increased with the increasing temperature, albeit at nanometer scale (Table S2 in our original Supplementary Information) and often unnoticeable by naked eyes. As suggested by the Reviewer, the temperature-induced roughness may be caused by the grains growth, plus the thermal expansion mismatches between the grains and grain boundaries. As the Reviewer might have also overlooked, we have indeed displayed data for adhesion to other surfaces (please see: Fig. 3d and the associated discussion in the top paragraph of Page 10), including a FEP film, aluminum foil, silicon wafer. Similar*

thermally-induced adhesion enhancements as observed with the copper foil were seen for our CNT dry adhesives against a FEP film and aluminum foil, but not for silicon wafer (a temperature-insensitive hard smooth surface). These results are in a good agreement with the newly-developed nano-interlocking model (Eq. S10 in Page 9 and the associated discussions in our original main text and Supplementary Information), confirming the contribution of the nano-interlock interactions, in addition to the vdW force, to the observed thermally enhanced adhesion, rather than just the connection between copper and the tubes only that gives the enhancements.

To address the Reviewer's concern, we have added the instrumental type of AFM and the roughness measurement details in the METHODS (please see: Page 14 of our revised main text), along with the newly obtained temperature-dependent AFM images for FEP, Al and Si wafer in Fig. S8(b) in our revised Supplementary Information.

2. English and Grammar: many improvements and corrections are needed but predominantly visible is the typo in the title which should say 'range' rather than 'rang'.

Response: *We thank the Reviewer for his/her careful reading. We have corrected the typos and gramma errors. A native English speaking person has proofread our revised manuscript.*

3. In the abstract 'folds' need to be corrected to 'fold' and 'behavior' should be corrected throughout to 'behaviour'. Correct the following sentence to: '...the highest adhesion strength among all known dry adhesives) over a temperature range of -196 ~ 1000°C.' Also, correct the '...this unusual behaviour led...' to: '... This unusual adhesion behaviour leads...'. Correct 'double-side' to 'double-sided'.

Response: *Once again, we thank the Reviewer for his/her careful reading, and have made all the required corrections accordingly.*

4. On Pg3, '...an extremely high temperature (e.g., 1000 °C...). The melting point of Cu is 1085°C so the copper is nearly molten!?

Response: *We thank the Reviewer for your careful reading. Yes, we made the test up to 1000 °C, which is close to the melting point of Cu (1085°C), but the copper was not molten, to induce high surface roughness, and hence the highly enhanced adhesion strength.*

5. On Pg4, '...the highest adhesion strength among all known CNT dry adhesives!'.

Isn't this just a repeat of what was said above?

Response: *We thank the Reviewer and agree with his/her point of view. Although it is repeated what said in the Abstract, we think it is better to be kept here (at the end of the first Paragraph on Page 4) as it is the only one-time appearance in the text. If the Editor insists, however, we are happy to remove it.*

6. On Pg6, '... The in-situ and ex-situ temperature controlled measurements shows agreements in the adhesion enhancement trend, implying the irreversible...'. Is it really irreversible? Explain and prove.

Response: *We thank the Reviewer for his/her comment. As reported in our manuscript, the CNT adhesion against thermally roughed surfaces consists of the vdW force and nano-interlocking interaction. While the van der Waal (vdW) interaction is reversible, the nano-interlocking interaction is irreversible as also confirmed by the nano-interlock model and our SEM images (Eq. S10 and the associated discussions as well as Figs. 2 and S15 in our original manuscript). Whenever the enhancement factor (defined as the normalization of adhesion strength at a specific temperature to the corresponding adhesion strength at room temperature – please see: Lines 10 – 11 on Page 8 in our original manuscript) is greater than 1, therefore, the adhesion becomes irreversible whereas CNT adhesions against to smooth surfaces at room temperature are mainly the vdW force in nature that is reversible. To address the Reviewer's concern, we have modified the sentence into “The in-situ and ex-situ temperature controlled measurements show an agreement in the adhesion enhancement trend, implying that the irreversible adhesion enhancement with increasing temperature ($\geq \sim 200^\circ\text{C}$) has been reliably measured.”*

7. Pg6, Needs to be corrected to: '...was the result from a newly-discovered "nano-interlock" dry adhesion mechanism. To elucidate the "nano-interlock" dry adhesion mechanism, we took SEM images of the CNT...'

Response: *We agree with the Reviewer's point of view and have modified the sentence accordingly.*

8. Pg7, Correct 'stability'...

Response: *We thank the Reviewer and have corrected the typo.*

9. Pg8, 'Clearly, therefore, an unknown yet decisive factor, in additional to the vdW and viscoelastic interactions...'. Correct this to: 'Clearly, therefore, an unknown yet

decisive factor, in addition to the vdW and viscoelastic interactions...'

Response: *We thank the Reviewer for his careful reading and have made the correction accordingly.*

10. Pg10, Correct 'As can be seen from the above discussion, our CNT dry adhesives hold potentials for various..' to: '...As can be seen from the above discussion, our CNT dry adhesives have potential for various...'

Response: *As suggested by the Reviewer, we have made the modification.*

11. Pg11, Should be 'data' not 'date'?

Response: *Once again, we thank the Reviewer for his/her careful reading and have made the correction accordingly.*

Reviewer #2 (Remarks to the Author):

In this paper the authors measure the temperature dependence of the adhesion of carbon nanotube dry adhesives. They find that the adhesion of these dry adhesives shows an increase in strength with temperature approaching ~ 140 N/cm². They propose an "interlock" model to account for this behavior. The ideas are novel but the quality of the data and the presentation needs some work. These are outlined below. I support publication of this manuscript after a few comments are addressed.

Response: *We thank the Reviewer for his/her recognition of the novelty of our work and for the nice recommendation.*

1. Did the authors try heating up and cooling and then measuring the adhesion after cycling the temperature? Does this annealing of the contact change the adhesive properties?

Response: *We thank the Reviewer for his/her insightful comments. As reported in our original manuscript, the temperature-dependence of the VA-DWNT adhesion had been measured from the CNT dry adhesives that were subjected to thermal treatments under both in-situ (heated and adhesion measurement in TA Instruments, RSA-G2 with Environmental Test Chamber) and ex-situ (heated inside of a muffle furnace and cooled down to room temperature for the adhesion force measurements) temperature control (Fig. S5 in our original Supplementary Information). In the ex-situ*

temperature controlled measurements, the samples were heated using a muffle furnace (LABEC 240V/2000W, Ceramic Engineering), and then cooled down to the room temperature for measurements. The in-situ and ex-situ temperature controlled measurements showed the same adhesion enhancement trend (please see: the first Paragraph on Page 6 in our original manuscript).

2. Grammatical correction on page 5 "consistent with the previous report"

Response: *We thank the Reviewer for his/her careful reading and have made the grammatical correction accordingly.*

3. Grammatical correction on page 6 "furnace and cooled down"

Response: *Once again, we thank the Reviewer for his/her careful reading and have made the grammatical correction accordingly.*

4. Grammatical correction on page 6 "(Fig. 1n) resulted from a newly-discovered"

Response: *As suggested by the Reviewer, we have made the correction.*

5. Typo on page 7 " thermal stability"

Response: *We thank the Reviewer for his/her careful reading and have corrected the typo.*

6: Typo on title "Temperature Range"

Response: *We thank the Reviewer for his/her careful reading and have corrected the typo.*

7. Please explain what the error bars in the figures represent. How were they calculated? Figure 1n has error bars and also Figures S4f. For the case of Fig. S4F are those variations between samples and how many samples?

Response: *We thank the Reviewer for his/her comments. In Figure 1n and Figure S4f, every adhesion strength data was averaged based on five samples tested under the same conditions with the whole range of the measured values from each of the five sample groups covered by the corresponding error bar. For the case of Fig. S4f, we*

have five samples subjected to plasma etching under the same conductions for each of the date points corresponding to different etching durations. The variation of the error bars seen in Fig.S4f is most probably related to the roughness mismatch between the substrate and the plasma-etched CNT adhesives with the longer error bar for the bigger roughness mismatch.

8/9. Typo on Supp Fig. S14 caption "ambient temperature"

Response: *Once again, we thank the Reviewer for his/her very careful reading and have corrected the typo.*

10. Typo on Supp Fig. S15 caption "weight was held on the painted wall"

Response: *Once again, we thank the Reviewer for his/her very careful reading and have corrected the typos.*

11. Typo on Supp Fig. S15 caption "after deliberately taking each"

Response: *Once again, we thank the Reviewer for his/her very careful reading and have corrected the typo.*

12. Please specify how the adhesion strength was calculated from the measurement? Also, how did the adhesion strength vary with contact area.

Response: *The adhesion strength was calculated by dividing the value of the adhesion force by the CNT adhesive size (i.e., the contact area). Figure R1 shows the relationship between the adhesion strength and the contact side length of square CNT adhesive samples (i.e. the CNT adhesive contact area). As can be seen in Figure R1, the adhesion strength unchanged with changing the adhesive contact area up to $1 \times 1 \text{ cm}^2$, followed by a slight decrease in the adhesion strength with further increasing the adhesive contact area up to $2.5 \times 2.5 \text{ cm}^2$, presumably because the contact is normally poorer with a larger contact area for all sorts of dry adhesives.*

Figure R1. The relationship between the adhesion strength and the contact side length of square CNT adhesive samples (i.e., CNT adhesive size).

13. The images in Fig. S4a - e look almost identical. I don't see any nanotube bundling as in Figure 1 f. Why is this?

Response: We thank the Reviewer for his/her insightful comment. Although the view angle and magnification for Figs. S4 and 1f are somewhat different while there are also some sample variations, the general trends observed are similar. To address the Reviewer's concern, we performed FE-SEM imaging (Figure R2) under a relatively higher magnification (10K) for a plasma-etched CNT array. As can be seen in Figure R2, the plasma etching initially removed the nonaligned nanotube segments with the concomitant top node formation to cause the nanotube bundling. However, over etching could cause the collapse of the nanotube bundle with a decreased adhesion strength.

Figure R2. FE-SEM images (10K magnification) showing the effect of plasma etching on the CNT array structure.

Reviewer #3 (Remarks to the Author):

The authors present a carbon nanotube based dry adhesive that increases in adhesion strength as temperature increases. A mechanism is proposed where the nanotubes 'screw' into naturally or temperature-induced rough surfaces enabling the increase in adhesion as temperature increases. The work goes on to show that these materials and mechanisms can also be useful for electrical and thermal conduction at interfaces.

Although interesting, the experimental results and procedures, novelty, and claims do not match the requirements for publication in Nature Communications. Specifically:

1) The experimental procedure for measuring the adhesive strength is not standardized and does not appear to be precise. Specifically, the adhesives were evaluated by pulling on a force scale as a butane torch was aimed at the interface. Without a controlled application of load and heat the results could contain artifacts (from rate effects, non-controlled loading conditions, localization of heat, etc.) and may not represent the actual interfacial strength.

Response: *We thank the Reviewer for his/her comments. As mentioned in our original manuscript, we used the torch heating to just make a quick demonstration (please see: Line -6 on Page 4 in our original manuscript) for the possible use of our CNT dry adhesives over a wide range of temperatures (Figs. 1h-j). Nevertheless, the temperature-dependence of the VA-DWNT adhesion demonstrated with the butane torch was confirmed by more careful adhesion strength measurements from the CNT dry adhesives subjected to thermal treatments under both an in-situ (heated and adhesion measurement in TA Instruments, RSA-G2 with Environmental Test Chamber) and ex-situ (heated inside of a muffle furnace and cooled down to room temperature for the adhesion force measurements) temperature control (please see: Fig. S5 in our original Supplementary Information). As also stated in our original manuscript, “the above (torch heating) observed temperature-dependence over the extremely wide temperature range from about -190.7 to 1033 °C was further confirmed by similar adhesion forces measured from the CNT dry adhesive being thermally treated with an in-situ (TA Instruments, RSA-G2 with Environmental Test Chamber) or ex-situ (heated inside of a muffle furnace and cooled down to room temperature for the adhesion force measurements) temperature control” (please see: the end of Paragraph 1 on Page 6 in our original manuscript). These two methods involved temperature controlled measurements and produced the same adhesion enhancement trend (please see: Fig. S5 in the first Paragraph on Page 6 in our original manuscript), leading to the reliable adhesion strength measurements. Even in the case with the torch heating for the quick demonstration, a thermal couple was placed close to the adhesive for displaying the adhesive temperature through a temperature reader, as shown in Fig. S1 in our original manuscript. As such, we believe our data are reliable, though there is no standard method/equipment currently available for such measurements.*

2) The work claims to present the highest adhesion strength for any carbon nanotube adhesive, and more broadly, any dry adhesive. This is inaccurate. Measurements on the adhesion strength of a single gecko seta demonstrated values on the order of 400 N/cm² (Autumn et al. Nature 2000). Therefore, this is not the strongest dry adhesive. Additionally, it is not the strongest carbon nanotube based adhesive, as Rong et al. (Adv. Mat. 2014) demonstrated a CNT adhesive with a shear adhesion strength of 185 +/- 50 N/cm².

Response: We thank the Reviewer for his/her careful reading and for sharing the aforementioned publications with us. As the Reviewer may be aware, however, the adhesion strength of a given dry adhesive sample depends also strongly on the contact situation. Compared to gecko feet, a single gecko seta can show a much better contact with a substrate, and hence a much higher adhesion strength. While the value of 400 N/cm^2 was measured for a single gecko seta, the first sentence of the above-mentioned Autumn et al. Nature 2000 paper (i.e., Autumn, K. et al., Nature **2000**, 405, 681) states: “The foot of a Tokay gecko (Gekko gecko) has about 5,000 setae mm^{-2} (ref. 4) and can produce 10N of adhesive force with approximately 100 mm^2 of pad area”. Among the dry adhesion community, therefore, 10 N/cm^2 is the well-acknowledged adhesion strength for Gecko feet pad. In Rong’s paper, the hierarchical pillar array (HPA, i.e., a square array of SU8 epoxy pillars) is coated with carbon nanotube forests (CNTFs) on the top of each of the SU8 epoxy pillars. In the first sentence in Page 1459 of the above-mentioned 2014 Adv. Mat. Paper (i.e., Rong, Z. et. al., Adv. Mater. **2014**, 26, 1456), Rong et al. state: “The HPA-covered surface with f (length/diameter) = 10 showed a shear stress of $\sigma_c = 185 \pm 50 \text{ N/cm}^2$ (Figure 4a) normalized to the actual contact area, which is higher by nearly one order of magnitude compared to the pure CNTF substrate.”, while the actual adhesion strength of the HPA material has been described as “This yields a shear stress of $7.8 \pm 1.2 \text{ N/cm}^2$ for the $f = 10$ HPA. While the maximum shear force of the $f=10$ HPA is comparable to that of CNTF, it has a nanotube area coverage of less than 5%” (5th – 8th line of the fourth paragraph in Page 1458 of 2014 Adv. Mat. paper). Therefore, the adhesion strength for the non-patterned CNTF should be $7.8 \pm 1.2 \text{ N/cm}^2$ to $18.5 \pm 50 \text{ N/cm}^2$, and the adhesion strength (143 N/cm^2) reported in our manuscript is still the highest adhesion strength among all known pure CNT dry adhesives. Besides, the value of $185 \pm 50 \text{ N/cm}^2$ reported in Rong et al. (Adv. Mat. **2014**, 26, 1456) paper was obtained by taking account only the CNTF area, which is less than 5% the actual HPA substrate area, leading to an far overestimated adhesion strength ($185 \pm 50 \text{ N/cm}^2$). Thus, this value was not listed in our Table S1, although the paper was cited as Ref. 5 in our original manuscript. To address the Reviewer’s concern, however, we have changed the phase of “all know CNT dry adhesives” into “all know pure CNT dry adhesives” in our revised manuscript in order.

3) The "nano-interlock" mechanism is not sufficiently supported. The schematics are clear enough to show the reader the hypothesis, however it is unclear how the images relate to the schematics. Without further evidence the current presentation is not convincing. Additionally, it is not clear how this 'could be applied to the development of various new dry adhesives and novel features.' It appears to be specific to this material set under the examined conditions.

Response: We thank the Reviewer for his/her insightful comments. As shown in Figs. 3 and S10 and associated discussions in our original manuscript, the individual CNTs

penetrated into the surface cavities like a “screw” (indicated by the red arrow in Fig. S10b) and corrugating along the surface profile after the web-like deformation. A model based on the above observation agreed well with the experimental data (please see: Figure S10 and associated discussion on Page 7-9 in Supplementary Information and Page 9 in the original manuscript). Therefore, the “nano-interlocking” mechanism is supported reasonably well by our experimental evidence and theoretical modelling, though it is challenging to directly image the “nano-interlock” structure due to tremendous technical difficulties. This paper is focused on communicating the thermally-enhanced adhesion of VA-CNTs over a wide operational temperatures and its mechanistic understanding, which is counter-intuitive with any conventional adhesive materials and had never been reported. Because of its novelty, we feel that our paper will spur unforeseen interests by other groups worldwide for further studies to further confirm the “nano-interlocking” mechanism from different angles and to apply the newly-developed strategy to other dry adhesives and substrates. As such, it is inappropriate to over emphasize on the detailed elucidation/imaging of the “nano-interlocking” mechanism/schematics, which could be the subjects for follow up studies.

Fig. 3d in our original manuscript shows the adhesion measurements on various substrates. The good agreement between the model calculation and experimental data for the silicon plate shown in Fig. 3d further confirms that the temperature-insensitive, hard, and smooth silicon surface could not support the formation of the thermally-induced interlocking structure. Besides, the good agreements between the model and experimental data shown in Fig. 3d for the FEP and Al surfaces indicate the absence of the CNT pull-out effect as the relatively low measuring temperatures (limited by the melting temperatures in these particular cases) cannot ensure a sufficient adhesion force with the target surface to detach CNTs from the CNT dry adhesive array in these particular cases (Please see: the first Paragraph on Page 10 in our original manuscript). These results clearly indicate the thermally-induced “nano-interlock” interactions described above can be applied to various substrate surfaces over a large range of temperatures. Thus, the newly-discovered node-guided nano-interlocking adhesion mechanism can be used for the development of various high-performance CNT dry adhesives against various substrate surfaces, including rough metal foils, plastic films, and even painted walls, as exemplified in this manuscript. To address the Reviewer’s concern, however, we have added “CNT” between the “development of various high-performance” and dry adhesives” in the last sentence of our revised manuscript (please see: the end of the main text on Page 12 in our revised manuscript).

4) The introduction discusses dry adhesives inspired by the gecko. One of the more intriguing aspects of the gecko is the highly reversible nature of the adhesive, where the adhesive can be applied and removed over many cycles. The CNT adhesive in this work is not demonstrated over multiple loading and release cycles, and upon removal it appears to damage the surfaces to which it was attached, as demonstrated in Figure S15. This adhesive is therefore more suitable as a permanent adhesive, and should be compared to other permanent adhesives instead of the pressure sensitive varieties such as 3M's double sided and thermally conductive tape. A more relevant and compelling comparison would be against permanent adhesives. In this case, the CNT adhesive is not as strong as established adhesive technologies such as epoxies which can obtain strengths on the order of 1500 N/cm^2 .

Response: *We thank the Reviewer for his/her interesting comments. As the Reviewer may be aware, the two most important features of the gecko adhesive are the dryness (thus, called dry adhesive) and the use of van der Waals (vdW) forces to mediate the adhesion, leading to the highly reversible nature of the gecko adhesive mentioned by the Reviewer. The vdW interaction makes the gecko adhesion to also depend on the pre-pressure (loading) – i.e., the contact situation. At the fundamental level, our CNT adhesive also possesses these three salient features governing the gecko adhesion. Actually, the damaged surfaces shown in Figure S15 in our manuscript indicate our CNT dry adhesive has a much stronger adhesion strength than that of a gecko foot due to the large-area contact, and hence strong vdW interaction, with the rough surface. Because of the fundamental similarities of our CNT dry adhesion and the gecko adhesion, we called our CNT adhesives as gecko-inspired dry adhesives - a term that has already widely used by the dry adhesive community for CNT and polymer arrays. If the Editor insists, however, we can consider to remove the phrases of “gecko-inspired” or the like, which, however, does not affect the importance and novelty of the results to be reported in this manuscript.*

In view of the fact that our CNT and gecko adhesives are fundamentally same in terms of the adhesion nature and the pre-pressure/loading effect, and that the irreversibility of our CNT adhesives indicates strong dry adhesion based on the van der Waals interaction - possibly in conjugation with the dry “nano interlocking” discovered in this study, we believe it is more appropriate to compare our CNT dry adhesives with the pressure sensitive 3M adhesive and thermally conductive tape (for studying the temperature-dependent adhesion behavior – please see the top of Page 6 of our original manuscript for some more reasons) than other permanent adhesives, such as epoxies that are based on the wet chemical crosslinking interaction rather than the vdW force.

5) The work appears very similar to previous publications from the research group,

specifically Qu, L., Dai, L. Adv. Mat 2007 ("Gecko-Foot-Mimetic Aligned Single-Walled Carbon Nanotube Dry Adhesives with Unique Electrical and Thermal Properties"). This reduces the novelty of the current contribution. Furthermore, these related works raise the question on the necessity of the multiwall carbon nanotubes (VA-MCNTs) presented in this work relative to single wall carbon nanotubes (VA-SWNT) presented in the previous work. In the author's previous work, adhesion strengths of 15-30 N/cm² were achieved with the SW-CNT at room temperature. This is similar to the 30-40 N/cm² reported here for VA-MCNTs. This leaves the reader wondering if the SW-CNTs have the same temperature dependent properties. In the larger scope, clarifying which CNT structure is most appropriate could help inform future research.

Response: *We thank the Reviewer for his/her careful reading and important comments regarding the direction for future research. As the Reviewer mentioned, we have previously reported "Gecko-Foot-Mimetic Aligned Single-Walled Carbon Nanotube Dry Adhesives with Unique Electrical and Thermal Properties" (Qu, L., Dai, L. Adv. Mat. 2007, 19, 3844), which, along with other relevant papers, has been cited as Ref. 7 in our original manuscript (please see the bottom of the first paragraph on Page 3 in our original manuscript). In the early work, however, the adhesion measurements were done between room temperature and ca.200 °C against the hard (ITO) glass slides, and hence the temperature-enhanced adhesion was not discovered (cf. Figure 3 in the present manuscript). As indicated by its title and clearly mentioned in the text, the present paper is focused on communicating the counter-intuitive behavior of carbon nanotube (CNT) dry adhesives to show a temperature-enhanced adhesion strength over a wide operational temperatures, together with the thermally enhanced electrical and thermal transports through the CNT dry adhesive. These never-shown-before adhesion and electrical/thermal performance was attributed to the rationally designed CNT strands on the adhesive top, leading to the newly-developed nano-interlock adhesion mechanism. Therefore, the present manuscript is novel and important, which is fundamentally different from our previously published Adv. Mater. (2007, 19, 3844) paper and all other publications. We are confident that this paper is of great broad interest to the readers of Nature Communications.*

In order to address the Reviewer's concern on the appropriate CNT structure(s) for the thermally enhanced adhesion, we have performed investigation on the effect of the nanotube wall number on the thermal behavior of the CNT adhesion. As can be seen in Figure R below, all the CNT adhesives, including the vertically-aligned single-walled, double-walled and multi-walled carbon nanotube arrays, showed the temperature enhanced adhesion with a similar trend. However, the multi-walled CNT

with a diameter exceeded 10 nm showed a very weak thermal enhancement of the adhesion. This is because the VA-MWNTs are too stiff to be collapsed to form the “nano interlock” structure whereas VA-SWNTs are too soft to efficiently penetrate into the surface cavities. Thus, the vertically-aligned double-walled CNT arrays are most appropriate for this study, as demonstrated in the present work (please see: the bottom of Page 7 and top of Page 8 in our original manuscript).

Figure R. a. Comparison of the temperature-dependence of the adhesion strength for the single-walled, doubled-walled and multi-walled CNT dry adhesive; b. TEM images showing the tube diameter and wall number for the samples tested, including SWNT, DWNT and MWNT (from left to right).

REVIEWERS' COMMENTS:

Reviewer #1 was not available to comment, but Reviewer #3 believes the technical points have been addressed. To further support some of the claims questioned by #1, #3 asks that citations be added to the manuscript for the roughening of the copper upon heating.

Reviewer #2 (Remarks to the Author):

I support publication after the below comments are addressed.

1. Response to 7: The description of what the error bars represents needs to be in the main text.
2. Response to 12: The description needs to be addressed in the main text. Please include Figure R1 in the supp. Info and rescale the y axis so that the white space above 50 is eliminated and the trend can be better observed.
3. Response to 13: Please include R2 in the supp. Info.
4. I agree with reviewer 3 comments that it is inappropriate to say "highest". I would get rid of claims of the "highest reported". You could just say among the highest or high but it is very difficult to prove that this is the highest since even a thorough literature review may miss something.

Reviewer #3 (Remarks to the Author):

The authors address many of the reviewer's points. However, a few points still need further attention as specified below:

- The authors should further address Reviewer 2, comment 7 regarding error bar concerns. The number of samples and what the error bar represents still do not appear in the figure legend.
- To further address Reviewer 2 comment 11, the authors should include figure R1 into the manuscript or supporting information and add a sentence to describe the decrease in the manuscript. Due to the decrease in adhesion shown in R1, the contact size for the 143 N/cm² value should be included in the abstract to provide more context to the reader.
- To further address Reviewer 3 comment 1, the authors should state in the text how the data in the figures are measured and calculated.
- The authors address the first half of Reviewer 3 comment 4 well, and the gecko-inspired nature of the adhesive is sufficiently supported. However, the CNT acts like a permanent adhesive as a release mechanism and reuse is not discussed or demonstrated, therefore comparison should still be made to permanent adhesives in the manuscript (literature/data sheet values would be sufficient, experiments do not necessarily have to be rerun).

Point-To-Point Responses To Reviewers' Comments (NCOMMS-16-10739-R2):

Reviewer #2 (Remarks to the Author):

I support publication after the below comments are addressed.

***Response:** We thank the Reviewer for his/her kind recommendation.*

1. Response to 7: The description of what the error bars represents needs to be in the main text.

***Response:** As suggested by the Reviewer, we have added specific descriptions for the error bars into the figure captions for figures with error bars (Figure 1&3, Supplementary Figures 3, 6, 7, 11&17 in our revised manuscript).*

2. Response to 12: The description needs to be addressed in the main text. Please include Figure R1 in the supp. Info and rescale the y axis so that the white space above 50 is eliminated and the trend can be better observed.

***Response:** We thank the Reviewer for his/her careful reading. As suggested by the Reviewer, we have modified Figure R1 to remove the white space above 50 and added the revised figure into the revised Supplementary as Supplementary Figure 6. We have also added the description on adhesion calculation under “Adhesion Strength Calculation” in the METHOD section of our revised manuscript.*

3. Response to 13: Please include R2 in the supp. Info.

***Response:** We thank the Reviewer for his/her comment and have added R2 with associated description into the revised Supplementary as Supplementary Figure 4.*

4. I agree with reviewer 3 comments that it is inappropriate to say “highest”. I would get rid of claims of the “highest reported”. You could just say among the highest or high but it is very difficult to prove that this is the highest since even a thorough literature review may miss something.

***Response:** We agree the Reviewer’s point of view and have changed the “highest” into “among the highest” throughout the revised manuscript.*

Reviewer #3 (Remarks to the Author):

The authors address many of the reviewer's points. However, a few points still need further attention as specified below:

-The authors should further address Reviewer 2, comment 7 regarding error bar concerns. The number of samples and what the error bar represents still do not appear in the figure legend.

***Response:** As suggested by the Reviewer, we have added specific descriptions for the error bars into the figure captions for figures with error bars (Figure 1&3, Supplementary Figures 3, 6, 7, 11&17 in our revised manuscript). We have added a specific description of what the error bars representing for each of the figures containing error bars (Figure 1&3, Supplementary Figures 3, 6, 7, 11&17).*

-To further address Reviewer 2 comment 11, the authors should include figure R1 into the manuscript or supporting information and add a sentence to describe the decrease in the manuscript. Due to the decrease in adhesion shown in R1, the contact size for the 143 N/cm² value should be included in the abstract to provide more context to the reader.

***Response:** We thank the Reviewer for his/her comments. As suggested by the Reviewer, we have added R1 into the revised Supplementary as Supplementary Figure 6. We have also added the description on adhesion calculation under "Adhesion Strength Calculation" in the METHOD section of our revised manuscript, along with the sample size for the measured adhesion strength value of 143 N/cm² in the revised abstract.*

-To further address Reviewer 3 comment 1, the authors should state in the text how the data in the figures are measured and calculated.

***Response:** We thank the Reviewer for his/her comment. Apart from the section of "Adhesion Force Measurements" in our original manuscript, we have added a new section entitled the "Adhesion Strength Calculation" in METHOD, along with specific descriptions for the error bars into the figure captions for figures with error bars (Figure 1&3, Supplementary Figures 3, 6, 7, 11&17 in our revised manuscript).*

-The authors address the first half of Reviewer 3 comment 4 well, and the gecko-inspired nature of the adhesive is sufficiently supported. However, the CNT acts like a permanent adhesive as a release mechanism and reuse is not discussed or demonstrated, therefore comparison should still be made to permanent adhesives in the manuscript (literature/data sheet values would be sufficient, experiments do not necessarily have to be rerun).

***Response:** We thank the Reviewer for his/her kind comments. As suggested by the*

Reviewer, we have added the following Supplementary Table 3 and associated discussions in the revised Supplementary for comparing our CNT adhesive with epoxy permanent adhesives.

Adhesive Type	Attachment Mode	Adhesion strength (N/cm ²)	Repeatability	Attachment Process	Temperature Range (° C)	Target Surface	Electrical and thermal properties
CNT Adhesive	Dry Adhesion /vdW interaction	37~145	Repeatable; Reduced to 30% the 2 nd round	Pressure sensitive, preloading required	-196~ 1033	Versatile: glass, metal, polymer	Electrically and thermally conducting
Conventional Permanent Glues	Wet chemical crosslinking interaction	Up to ~2500	Not repeatable	Cure, heat/pressure required	< 100	Surface selective	None